# Training Ultrasound Image Classification Deep-Learning Algorithms for Pneumothorax Detection Using a Synthetic Tissue Phantom Apparatus

**DOI:** 10.3390/jimaging8090249

**Published:** 2022-09-11

**Authors:** Emily N. Boice, Sofia I. Hernandez Torres, Zechariah J. Knowlton, David Berard, Jose M. Gonzalez, Guy Avital, Eric J. Snider

**Affiliations:** 1U.S. Army Institute of Surgical Research, JBSA Fort Sam Houston, San Antonio, TX 78234, USA; 2Trauma & Combat Medicine Branch, Surgeon General’s Headquarters, Israel Defense Forces, Ramat-Gan 52620, Israel; 3Division of Anesthesia, Intensive Care & Pain Management, Tel-Aviv Sourasky Medical Center, Sackler Faculty of Medicine, Tel-Aviv University, Tel-Aviv 64239, Israel

**Keywords:** model development, automation, tissue phantom, pneumothorax, ultrasound, porcine, human, artificial intelligence, deep learning

## Abstract

Ultrasound (US) imaging is a critical tool in emergency and military medicine because of its portability and immediate nature. However, proper image interpretation requires skill, limiting its utility in remote applications for conditions such as pneumothorax (PTX) which requires rapid intervention. Artificial intelligence has the potential to automate ultrasound image analysis for various pathophysiological conditions. Training models require large data sets and a means of troubleshooting in real-time for ultrasound integration deployment, and they also require large animal models or clinical testing. Here, we detail the development of a dynamic synthetic tissue phantom model for PTX and its use in training image classification algorithms. The model comprises a synthetic gelatin phantom cast in a custom 3D-printed rib mold and a lung mimicking phantom. When compared to PTX images acquired in swine, images from the phantom were similar in both PTX negative and positive mimicking scenarios. We then used a deep learning image classification algorithm, which we previously developed for shrapnel detection, to accurately predict the presence of PTX in swine images by only training on phantom image sets, highlighting the utility for a tissue phantom for AI applications.

## 1. Introduction

Ultrasound imaging is broadly used in medical care. One critical application is in emergency medicine (EM), as the adoption of point-of-care ultrasonography (POCUS) in EM has been supported by The American College of Emergency Physicians (ACEP) [1] and the Society for Academic Emergency Medicine [2]. With this support of POCUS within EM, standard ultrasonography training was integrated as part of EM residency programs. Since ultrasonography inclusion in the EM environment in the 1990s, EM use of ultrasonography has continued to increase [3]. Ultrasonography guidelines by ACEP made EM use of ultrasonography viable for abdominal aorta, biliary urinary tract, cardiac, deep vein thrombosis, musculoskeletal, ocular, pregnancy, procedural, thoracic, trauma and soft tissue purposes [4,5]. 

The effectiveness of ultrasonography in the detection of pneumothoraxes (PTXs) as a diagnosis tool has been looked at in comparison to traditional detection techniques. Computed tomography scans are used for PTX diagnosis, but due to the cost, limited accessibility, long acquisition and processing time, and radiation concerns, chest radiographs are more commonly used. Chest radiography for the detection of PTXs is less accurate than chest ultrasonography, and chest radiography often fails to detect pneumothoraxes [6,7]. With novice US users, motion mode (M-mode) ultrasound has been shown to be able to more easily identify PTX compared to traditional brightness mode (B-mode) [8]. PTX detection must be accurate and quick to prevent complications, such as respiratory and/or circulatory compromise, leading to cardiac arrest [9].

To aid in the detection of injury or abnormalities seen in medical imaging, deep learning approaches have been utilized. In general, deep learning is a subset of machine learning that aims to mimic the way humans gain knowledge and make decisions from that knowledge. For instance, deep learning models have been developed for use in CT scans or X-rays for pulmonary image analysis to classify the lung lobes, detect and locate any lung abnormalities and segment areas of interest [10]. Chest X-rays have also been used as datasets for the detection of PTX using an object detection algorithm, and they produced favorable results on this more time consuming imaging modality [11]. In addition, image segmentation approaches have been used as an initial step for the detection and classification of lung cancer [12]. GoogLeNet was used to train chest X-rays coupled with image augmentation strategies such as histogram normalization for the classification of benign and malignant tumors [13]. 

In the context of US imaging, a number of different approaches have been taken, such as coupling deep learning classification, object detection or segmentation models, with raw signals from US images rather than the instrument’s processed visual display [14]. Using B-mode US images, InceptionV3 network trained with US images of lungs could correctly detect COVID-19 and pneumonia vs. healthy lung images with 89.1% accuracy [15]. Deep learning models have also been developed to assist with identifying abnormalities during US tasks requiring high levels of US expertise, such as fetal imaging during pregnancies [16]. A recent review on deep learning methodology in the medical ultrasound space have shown detailed construction of popular deep learning architectures and their role with a multitude of medical uses [17]. The further inclusion of ultrasonography specific modes, such as M-mode (still image motion over time), coupled with non-computationally intensive deep learning models, can increase the likelihood of incorporating automated PTX detection with ultrasonography hardware. 

As ultrasonography use in the EM environment increases, more methods to train residents will be required. Ultrasonography-compliant phantoms that are currently used can be costly [18], limiting the number of residents that can use these commercially available phantoms. In order to increase the ease of access for EM residents that may not have the funds for commercial phantoms, groups have created ultrasound-compliant, inexpensive and easy-to-make phantoms for a variety of uses such as soft tissue abscess identification [19], biopsies [20], vascular access [21] and lung phantoms that mimic a variety of complications, including collapsed lungs [22]. 

In addition, as more advanced deep learning models become available, larger datasets and troubleshooting capabilities will be needed for PTX detection. This is especially true as AI is integrated into ultrasound machines for real-time deployment, requiring an animal model or clinical databases to troubleshoot, validate and evaluate the model accuracy. In response, here we detail the development of an inexpensive, simple method to create a PTX US-compliant phantom for the primary purpose of the preliminary development and troubleshooting of deep learning algorithms. The details of constructing the phantom molds and creating the phantom components are highlighted as well as its echogenic appearance similarity to swine tissue. Finally, the phantom image datasets are utilized for training an image classification deep learning algorithm and are tested with swine images.

## 2. Materials and Methods

### 2.1. Custom Thoracic Phantom Mold Construction 

A two-piece mold was developed based on measurements of a portion of the human thoracic region, modeled using computer-aided design (CAD) software (Autodesk Inventor, San Rafael, CA, USA), and was fabricated using either stereolithographic printing with High Temp resin (Formlabs, Somerville, MA, USA) or filament deposition modeling with co-polyesters (Ultimaker, Utrecht, The Netherlands) (Figure 1A). The rectangular phantom mold had a length of 106.5 mm, a width of 76.2 mm and a depth of 27.8 mm. The mold contained four straight ribs with elliptical cross-sections of varying sizes positioned to lay flush against the bottom surface of the mold (Figure 1A). This provided three unique intercostal spaces per mold variant. A removable floor was produced to seal the mold for constructing the phantom and was subsequently removed for ultrasound image acquisition. In total, three models were produced with different rib sizes and intercostal space widths to demonstrate subject variability. The CAD files are supplied as a Appendix A.

### 2.2. Synthetic Phantom Preparations

After the 3D printed rib model was produced, the bottom lid was secured to the mold using sealing tape (McMaster-Carr, Elmhurst, IL, USA). An amount of 300 mL of 10% Clear Ballistic Gel (CBG) (Clear Ballistics, Greenville, SC, USA) with 0.25% *w*/*v* flour (HEB, San-Antonio, TX, USA) was used to model the thoracic soft tissue and was melted in an oven at 130 °C. To mimic features of the muscular layer surrounding the ribs, the CBG was poured in layers of varying thicknesses and was allowed to set at slightly altering angles. A thin coating of silicone oil (Millipore Sigma-Aldrich, St. Louis, MO, USA) was applied between each CBG layer to mimic muscle-like striations in the ultrasound image. Half of the mixture (150 mL) was added using this layering technique, and the remainder was poured at once, filling to the mold’s rim. The entire pouring process took approximately one hour and was repeated two more times for the subject variability setups utilizing the other fabricated molds.

To achieve baseline images of a normal breathing lung, a lung phantom was constructed using the same 10% CBG with 0.25% *w*/*v* flour mixture. A 400 mL volume of the mixture was prepared and melted in a 1 L beaker and was whisked with a spatula to create miniature bubbles for mimicking the air–fluid interfaces of the lung. This process was repeated two more times to obtain three subject variabilities for the lung phantoms. Lastly, a skin mimicking layer was produced using EcoFlex 10 silicone (Smooth-On, Macungie, PA, USA). A small amount was poured into a rectangular tub and was allowed to cure, forming a thin sheet (2–4 mm thick) that was removed and laid on top of the rib phantom. 

### 2.3. Synthetic Phantom Apparatus Setup and Ultrasound Image Collection 

The model was set up as shown in Figure 1B using Aquasonic Clear Ultrasound Gel (Parker Laboratories, Inc., Fairfield, NJ, USA) to prevent trapped air at the interface between the probe and skin, the skin and rib phantom, and the rib phantom and lung phantom. The skin and rib phantom were held in place with a ring stand, and the lung phantom was positioned atop the shaker (Orbit LS, Labnet International, Edison, NJ, USA) and was allowed to rotate, creating a motion plane between the lung and the chest wall (rib phantom), simulating respiration. An HF50 ultrasound probe (Fujifilm, Bothell, WA, USA) was held stationary above the skin while the ultrasound unit captured the images. Pictured in Figure 1 on the ultrasound unit screen is an M-mode image of the various thoracic tissue layers, pleural line and lung phantom. 

A Sonosite Edge Ultrasound system (Fujifilm, Bothell, WA, USA) was used to acquire all ultrasound images. “Pneumothorax negative” clips were obtained for each rib–lung pair in B-mode (6 s clips) for each intercostal space at 30, 20 and 10 rpm settings on the shaker plate to simulate different respiratory rates. During the B-mode imaging, the probe was tilted approximately −45° to +45° for added image variability. In addition, triplicate 5 s M-mode images were taken of the middle intercostal space of the model at the 20 rpm shaker setting. Negative images typically presented as a “sea-shore” or “sandy beach” appearance in M-mode [23]. Lastly, for “pneumothorax positive” image sets, a vertical separation was introduced between the rib phantom and lung phantom by adjusting the ring stand, and B-mode clips were captured for each intercostal space as before, including a single M-mode 5 s window for the central intercostal space, which typically presented as a “barcode” or “stratosphere” signal [23]. This process was repeated for each rib–lung setup.

### 2.4. Euthanized Porcine Pneumothorax Model

The thoracic cavity of a porcine subject that had been recently euthanized from an unrelated Institutional Animal Care and Use Committee approved protocol was used to collect pneumothorax negative and positive images. An endotracheal tube (McKesson Medical-Surgical, Irving, TX, USA) remained in place post-euthanasia and was connected to a Bag-Valve-Mask (EMS Safety Services, Eugene, OR, USA) for continued ventilation. Baseline B-mode and M-mode images were collected with manual ventilation at 12 breaths per minute. Pneumothorax was created in a method similar to the one described by Oveland et al., with an 8 Fr feeding tube (Covidien, Mansfield, MA, USA) used as the pleural catheter [24]. The feeding tube was connected to a three-way valve (Amazon, Seattle, WA, USA) and a 100 mL syringe (Becton, Dickinson and Company, Franklin Lakes, NJ, USA). An amount of 1000 mL of air was gradually insufflated into the pleural space. Manual ventilation was again initiated, and positive B-mode and M-mode images were collected from the 3rd–5th intercostal space in the mid-clavicular line. Images from a single animal were used for comparison and deep learning model testing. 

### 2.5. Image Processing and Augmentation 

Frames from the B-mode clips were extracted using a Ruby script that employed ffmpeg. Each 6 s clip yielded 60 individual frames. All images were then cropped to remove ultrasound settings information displayed in the exported files and were resized to 512 × 512 squares using the MATLAB (MathWorks, Natick, MA, USA) image batch processing application.

For the deep learning training setup, M-mode images were split into subsections to expand the training dataset (Figure 2). The 5 s M-mode window was split into 100 single second cropped sections using a rolling window to crop images. X,Y coordinates for the top corner, and width and height pixel coordinates for the one second window size were determined using FIJI/ImageJ [25,26]. MATLAB was used to move through the 5 s M-mode US image and to evenly create the 100 separate frames via a rolling window methodology. Prior to use with the deep learning model, all images were adjusted to a size of 512 × 512 × 3 to meet the input size for the DL network (Section 2.6). 

For some training applications (Figure 2), images were augmented or further pre-processed prior to training with increasing complexity. The initial level of processing had three random augmentation operations applied to images: X reflection, Y reflection and image scale/zoom of 0.8 to 1.2x (Figure 2). This augmentation step was only performed on training images. For some training situations, an additional level of image preprocessing was performed on all images (training, validation and testing) and was added to account for brightness differences between swine and phantom image sets. Image histograms were matched up using an “imhistmatch” command in MATLAB, with a reference swine baseline image (Figure 2). Briefly, the reference swine image’s histogram was split into 64 bins, and each image was adjusted to try to match the histogram profile in each bin. All swine and phantom images were adjusted to match up image histograms, followed by algorithm training. The third level of augmentation randomly adjusted brightness and contrast in all training images. These adjustments were random for each image, using the “jitterColorHSV” command in MATLAB with a ±0.5 scalar factor limit for brightness and a 0.7 to 1.4 scalar factor limit for contrast (Figure 2). This level of augmentation was performed separately for contrast and brightness, effectively doubling the image set size for training. 

### 2.6. Algorithm Training and Testing

We utilized a previously developed deep learning network for initial algorithm evaluation with the tissue phantom [27,28]. This algorithm was tuned for the image classification of shrapnel within ultrasound images—termed ShrapML—and was re-trained in this effort for use with PTX detection. Briefly, ShrapML used an image input size of 512 × 512 × 3 and passed images through six sparsely connected convolutional neural network layers paired with max pooling layers. Next, a dropout layer set at 36% was used to reduce overfitting, followed by a flatten layer to linearize the input for a 214-node fully connected layer. Lastly, a 2-node fully connected layer, a softmax layer, and a classification output layer were utilized sequentially to allow for binary classification as PTX positive or negative. 

Training and the use of ShrapML for this work was conducted in MATLAB using the deep learning toolbox. Phantom images were processed as described in Section 2.5, followed by splitting the image sets into 70%, 15% and 15% for training, validation and testing, respectively. An additional swine image set was used for blind testing. A total of up to 100 epochs were used for training with a validation patience setting of 5 epochs; therefore, training could end early if validation loss was not improving to avoid overfitting. A learning rate of 0.001 and a batch size of 30 was used during training, and the lowest validation loss epoch was selected for the model as opposed to the final epoch. All training was performed using an Alienware laptop (Miami, FL, USA) running Windows 10 Pro and an Intel Core i9-9980 (2.40 GHz, 8 core) processor with 16 GB RAM using an Nvidia GeForce RXT 2080 (8 GB VRAM). 

After training, test images were used for evaluating model performance. Testing was conducted for split image sets set aside from phantom images and swine M-mode image sets. For the different training image sets used, 3 to 5 independent models were trained and tested using different random seeds. Confusion matrices were created using GraphPad Prism 9 (San Diego, CA, USA) based on true positive, true negative, false positive and false negative prediction categories. In addition, accuracy, precision, recall, specificity and F1 scores were calculated for each model. 

## 3. Results

### 3.1. Ultrasound B-Mode Image Results for the Synthetic Phantom Apparatus 

The diagnosis of pneumothorax using ultrasound techniques typically involves the identification of landmarks in the ultrasound field (rib shadows) and the presence of sonographic signs (lung sliding or B-lines) [29]. The baseline B-mode images captured from the porcine subject provided a view of the two rib shadows surrounding the third intercostal space with muscle and superficial tissue above (Figure 3A). The lower portion of this panel featured the view of the lung. The pleural line was evident between the ribs and radiated hyperechoic discrete lines downward, known as B-lines. In real-time ultrasound acquisition, the B-lines moved synchronously with respiration, and lung sliding was apparent (data not shown). In the porcine subject post-injury, the B-mode image showed the same anatomy as the baseline image but lacked the presence of the B-lines (Figure 3C). The baseline B-mode images acquired using the synthetic phantom in the platform also featured the same anatomical features: rib shadows, muscle tissue, lung tissue and the pleural lining. The B-lines were also apparent in the synthetic model (Figure 3B). When the motion of the lung was halted in the platform, the B-mode image of the phantom lost the B-lines (Figure 3D).

### 3.2. M-Mode Comparison of Porcine and Synthetic Platforms

An alternative approach of diagnosing pneumothorax involves M-mode imaging, maintaining a stationary transducer, and observing the resulting ultrasound image over time. The M-mode images were captured at the same timepoints as the B-mode images shown above. When the M-mode line was centered in the pleural lining, the baseline porcine subject provided the typical “sea-shore” sign (Figure 4A) [23]. The post-injury porcine M-mode image lost this appearance and instead presented a “stratosphere sign”, indicating a lack of lung sliding (Figure 4B). The synthetic phantom produced similar “sandy beach” characteristics (Figure 4C) when motion was functional on the apparatus, and it produced the “barcode” when motion was halted (Figure 4D). 

### 3.3. Image Classifier Algorithm Training

Next, we evaluated how the image sets acquired from the PTX phantom apparatus could be deployed for deep learning algorithm development. We utilized a previously developed convolutional neural network for shrapnel detection in tissue—termed ShrapML—for the detection of PTX injuries in the phantom model. After processing M-mode images into segments (see method Section 2.6), the trained model was 100% accurate on classifying positive/negative PTX images from the synthetic phantom apparatus. For a more realistic testing approach, swine PTX M-mode images were evaluated by the model, resulting in 50% overall accuracy, with all model predictions being negative (Figure 5A). To reduce overfitting to the training data, zoom and flip image augmentations were included in the training process, but the overall accuracy remained at 50% (Figure 5B). Image brightness was noticeably different between swine and phantom images, so a function was applied to all images to equilibrate image brightness histograms to a reference swine image. This additional pre-processing step improved swine image accuracy to 96.9% true positive and 68% true negative, with an overall accuracy of 82% (Figure 5C). To further remove the brightness/contrast differences from the training set, images were augmented randomly for these two additional operations, resulting in 99.8% true positive and 87.3% true negative swine test accuracy, with an overall accuracy of 93.6% (Figure 5D). Overall, these augmentation operations improved accuracy from 50% without any augmentation to as high as 93.6% with multiple augmentation methods. Additional performance metrics for each model are shown in Table 1. 

## 4. Discussion

Ultrasound has a wide use case in emergency and military medicine due to its portability and range of potential applications. However, ultrasound imaging is only useful for emergency applications if images can be properly analyzed in a timely manner and predict correct diagnostics. It is for this reason that AI has the potential to make US more accessible at the point of injury and during emergency triage situations by automating the image interpretation process. We have previously highlighted this capability for shrapnel detection, and others have shown AI for US during a range of emergency applications, such as abdominal free fluids and COVID-19 [30,31,32,33,34]. Major challenges for AI, and specifically deep learning models for US image interpretation, are acquiring image databases for training or testing with phantoms or animal subjects for the real-time evaluation of DL models. Here, we detailed the creation of a PTX synthetic phantom apparatus and its utility for DL model training. 

The PTX phantom was simple to configure, relying on 3D-printed parts and synthetic ballistic gelatin. The resulting images were similar to swine images for both B-mode and M-mode ultrasound data captures. The phantoms were shelf stable, allowing for repeated use, and, by automating the lung motion, a wide range of respiratory rates was possible. Given the 3D-printed parts, gelatin fabrication method, and respiratory rate control, the phantom apparatus design can be tuned to the user’s needs or can be evaluated with modifications to simulate subject variability. Thus, a simple PTX phantom apparatus such as this can reduce the extent of swine or human image collection requirements, which may be needed for PTX imaging applications. It does not replace the need for live data collection, but initial troubleshooting, training, and algorithm development may be possible with a synthetic phantom apparatus. 

Focusing on the algorithm development application for the PTX phantom, an image classification algorithm was capable of distinguishing between positive and negative PTX images. Algorithm work was focused on M-mode images initially, as the time aspect for these images allowed for easier identification of PTX injury states and is more commonly used for PTX identification [8,29]. However, preliminary training with phantom B-mode images had a similar near-100% accuracy for phantom test data sets (data not shown). Although the accurate detection of PTX in phantom US images is evidence that the injuries are predictable in the synthetic phantom apparatus, that does not indicate that the PTX injury would be detectable in swine PTX image sets. Without image augmentation, the model could not make accurate predictions in swine, which we hypothesized was due to initial datasets overfitting the model due to the brightness and contrast differences between the positive and negative images, as opposed to the key features of the PTX. As a result, image augmentation in these brightness and contrast confounding factors resulted in 93.6% prediction accuracy when testing with swine images. It is worth noting that the strong training performance improvement with brightness or contrast adjustments by histogram normalization and augmentation steps, may mean that the other conventional augmentation steps, such as image zoom and flip, may not impact model performance. However, in this proof-of-concept study, changes to the training set were compounded, as opposed to being evaluated separately. The further optimization of image augmentation, deep learning network optimization, and larger swine and synthetic phantom apparatus datasets can likely improve on this prediction performance, but it provides evidence that the PTX phantom apparatus was similar enough to swine US images to make accurate predictions and, thus, supplement the need for swine or human data collection for PTX AI applications. 

Other applications for this PTX phantom setup include different pathophysiological conditions, real-time use cases, and additional anatomical designs. M-mode imaging has been useful in distinguishing cardiogenic pulmonary edema versus noncardiogenic alveolar interstitial syndrome [35] and in evaluating diaphragm movement [36,37,38]. These diagnoses can potentially be automated using this technology. Moreover, different designs can be configured for hemothorax, with blood in the pleural cavity instead of air, allowing for the evaluation of blood accumulation at different extended Focused Assessment with Sonography in Trauma (eFAST) scan points. Second, this phantom is well suited for troubleshooting PTX AI models and evaluation in real-time. For these applications, large image repositories cannot be substituted for live animal/human testing. These applications may include integration and debugging AI within US systems for real-time evaluation, AI focused on confirming that the US probe is aligned appropriately for proper eFAST image collection, and computer vision/robotic US data collection applications. Third, the current phantom apparatus is configured for the human anatomy but can be configured to other organisms’ anatomies to meet the end user’s application. 

There are a few limitations for the current study. First, the phantom is not suitable for intervention training or testing, such as needle thoracostomy, that may follow positive PTX diagnoses. Second, the algorithm for this use case was built for shrapnel identification as opposed to PTX identification. A more widely used architecture such as ResNet50 [39], VGG16 [40] or MobileNetv2 [41] may be preferred, or a Bayesian optimized network architecture for this specific application may further improve on performance [42,43]. Third, phantom similarities were only compared to PTX positive/negative image sets collected in post-mortem tissue from a single swine subject. Larger live swine or human PTX image sets will be needed to compare against the phantom to confirm the capabilities more widely for the phantom apparatus, as well as its algorithm training potential. Additionally, the use of controlled models of pneumothorax (precise volumes of air pockets) may enable better definition of the algorithm’s detection threshold. 

## 5. Conclusions

In conclusion, this synthetic phantom apparatus development addresses a current gap in the generation of large image datasets for deep learning algorithm development. Producing large quantities of ultrasound images with subject variability is essential for algorithm training and to ensure the data do not overfit the model. Typically, these have been generated from live animal or human imaging studies, and these experiments are extensive and have complicated regulatory processes. The synthetic phantom apparatus developed here is easy to produce, simple to modify to mimic different subjects, and can allow for virtually limitless image collection. For these reasons, we believe that the developed phantom apparatus will assist with automating diagnoses using artificial intelligence methodologies. 

## Figures and Tables

**Figure 1 jimaging-08-00249-f001:**
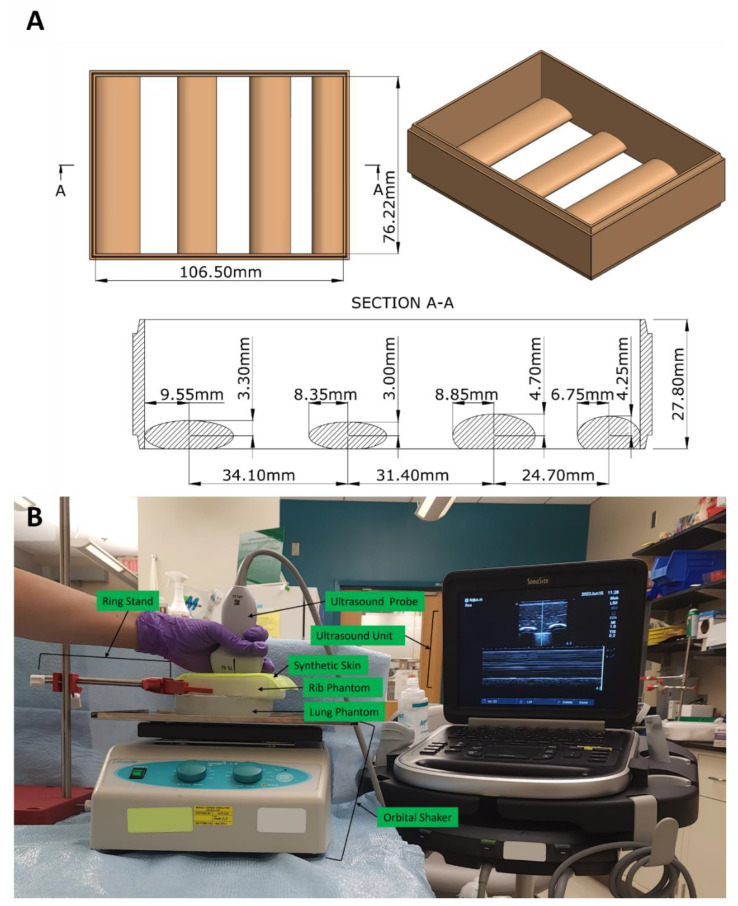
Synthetic platform for pneumothorax image acquisition: (**A**) CAD rendering of the thoracic cavity with a dimensioned cross-section view of one variation; (**B**) Image acquisition platform incorporating (top to bottom): ultrasound probe manually held in place and connected to the ultrasound unit, synthetic skin positioned atop the rib phantom that was held in place by the ring stand, and the lung phantom located on the bed of the orbital shaker.

**Figure 2 jimaging-08-00249-f002:**
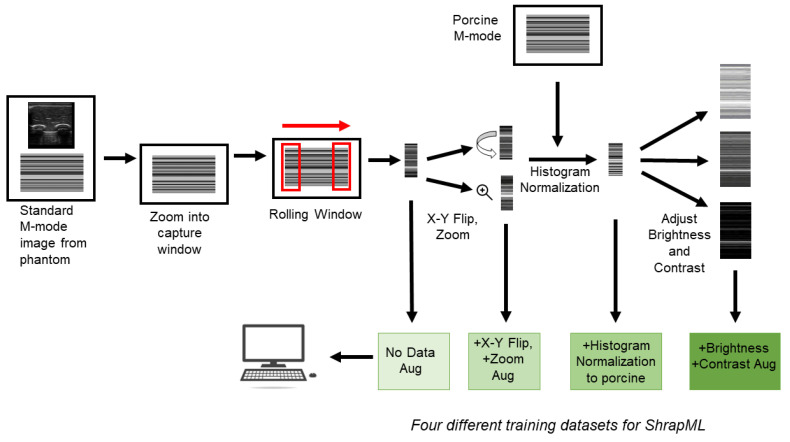
Image processing and algorithm training pathway. M-mode images were obtained from the synthetic phantom, and coordinates of the time capture window were identified. A rolling window method was performed to isolate 100 individual panels from each capture window. The individual panels were used as the initial training set (No Data Aug box). A basic augmentation step was performed to the original dataset to create the second training set (+X-Y Flip, +Zoom Aug box). The third image preprocessing step used a porcine M-mode image to normalize the histogram for the phantom panels (+Histogram Normalization box). The final augmentation step included brightness and contrast augmentation to generate additional images (+Brightness, +Contrast Aug box).

**Figure 3 jimaging-08-00249-f003:**
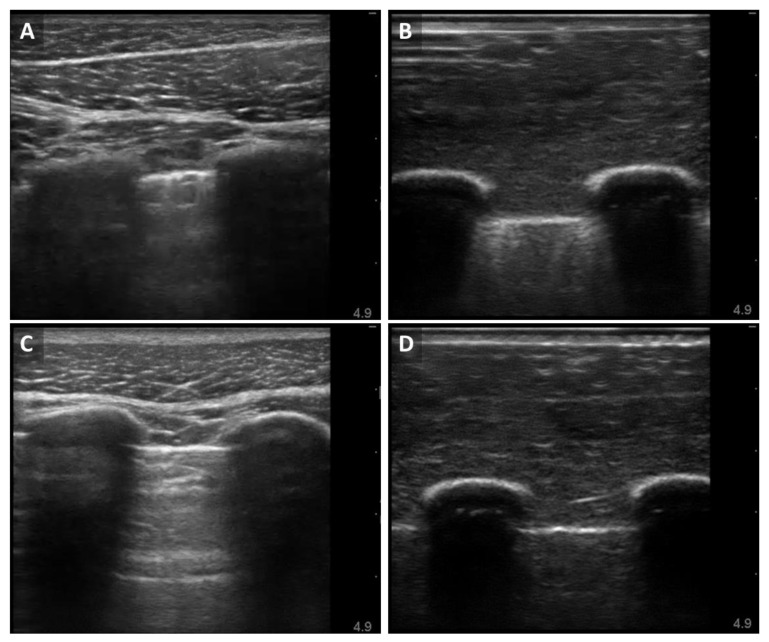
B-mode image comparison with euthanized porcine subjects and synthetic phantom: (**A**) Baseline thoracic image acquired from porcine subject; (**B**) Baseline image acquired from rib and lung phantom apparatus; (**C**) Post-injury image acquired from same porcine subject; (**D**) Pneumothorax positive image acquired from synthetic phantom apparatus.

**Figure 4 jimaging-08-00249-f004:**
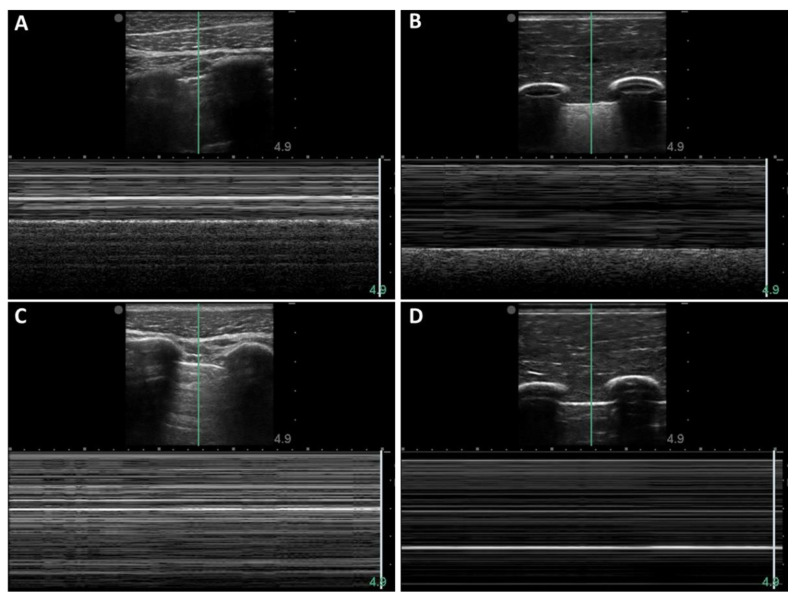
M-mode image comparison with porcine subjects and synthetic phantoms: (**A**) Baseline thoracic image acquired from porcine subject; (**B**) Baseline image acquired from synthetic phantom apparatus; (**C**) Post-injury image acquired from same porcine subject; (**D**) Pneumothorax positive image acquired from synthetic phantom apparatus.

**Figure 5 jimaging-08-00249-f005:**
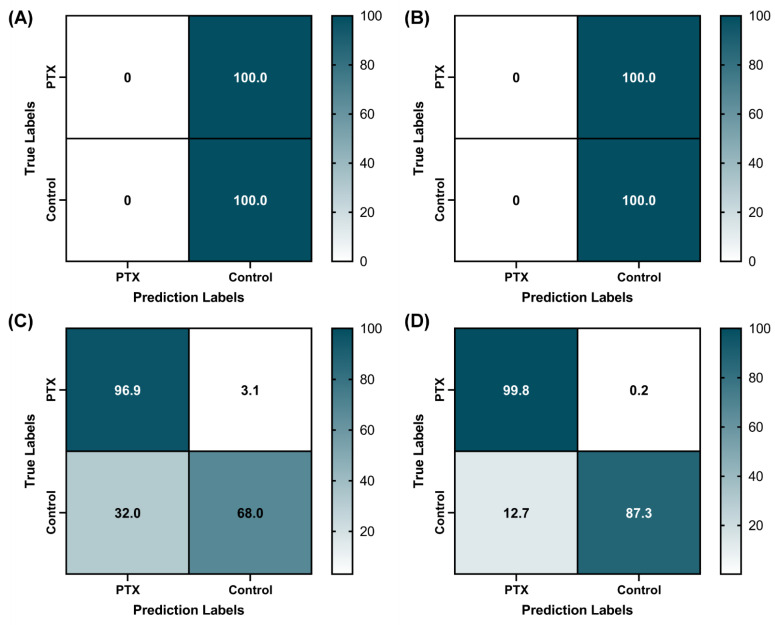
Confusion matrix results for swine image predictions using four image augmentation approaches. Average confusion matrix results are shown for (**A**) no image augmentation (*n* = 3 independent trained models), (**B**) using zoom and flip image augmentation (*n* = 3 trained models), (**C**) further including image histogram swine/phantom matching (*n* = 5 trained models), and (**D**) finally including contrast and brightness image augmentation (*n* = 5 trained models). All models were trained on processed M-mode segments collected in the synthetic phantom apparatus, and confusion matrix test results are shown for swine M-mode image sets (*n* = 400 total image segments, equal number of positive and negative PTX images). Results are shown as mean percent across replicate trained models for each confusion matrix category.

**Table 1 jimaging-08-00249-t001:** Summary of model performance metrics. Results are shown for each image augmentation methodology for phantom split test data sets and swine test image sets. Augmentation steps follow the pathway described in Figure 2.

Augmentation	Testing Type	Accuracy	Precision	Recall	Specificity	F1
None	Split	1.000	1.000	1.000	1.000	1.000
	Swine	0.500	-	-	1.000	-
Only X-Y Flip and Zoom	Split	1.000	1.000	1.000	1.000	1.000
	Swine	0.500	-	-	1.000	-
+ Histogram Normalization	Split	0.999	0.999	1.000	0.999	0.999
	Swine	0.825	0.792	0.969	0.68	0.862
+ Contrast/Brightness	Split	1.000	1.000	1.000	1.000	1.000
	Swine	0.936	0.898	0.998	0.873	0.942

## Data Availability

The datasets generated during and/or analyzed during the current study are available from the corresponding author upon reasonable request.

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
