# Peer review of "Training Ultrasound Image Classification Deep-Learning Algorithms for Pneumothorax Detection Using a Synthetic Tissue Phantom Apparatus"

_2313-433X, 2022, doi:10.3390/jimaging8090249_

Round 1

Reviewer 1 Report

CNN, one of the deep learning methods, is used in this paper to demonstrate the effectiveness of the proposed method by learning artificially created images of pneumothorax and testing them on images of real swine pneumothorax. Histogram normalization and augmentation by brightness and contrast seem to be effective. This makes it practical to quickly detect pneumothoraces.

The following questions are raised and would like to be explained or improved.

1        Make clear the reason for using M-mode images for analysis instead of B-mode.

2        It says average of confusion matrix, but it is not clear how many independently trials were used to calculate this.

3        Explain in detail how histogram normalization is performed.

4        Similarly, explain in detail how augmentation of brightness and contrast is performed.

5        From the experimental results (Figure 5 and Table 1), the augmentation of X- Y- Flips and Zoom may not be required at all.

Reviewer 2 Report

Summary:

This paper presents an AI solution for Pneumothorax Detection which has been trained using images obtained from a phantom. Ultrasound is a modality that lacks data for training AI-based methods, therefore the idea to use a phantom to generate a dataset for training can potentially eliminate the burden of data acquisition. 3 phantoms of slightly different rib dimensions were created to get data to train an already existing AI model. Data from a porcine model was used for testing. The authors claim that the method is able to classify the porcine images with high accuracy.

Major comments:

Literature is weak and should be extended to include relevant works in the literature. For instance, a summary of deep learning methods for ultrasound would have been useful.

I have concerns regarding the data split used for training, validation and testing.

Training: The authors state that phantom images were split into training, validation and testing, however this strategy did not consider images from the same phantom and acquisition can be in more than one dataset. Therefore, results on the “split testing” are not informative/valid as there may be a high correlation between the images used for training and testing. This is confirmed by Table 1.

Test: Images from a single animal were used for testing, which is not sufficient. Although I understand it may be difficult to obtain, human data would be preferable to demonstrate the method.

Results: Confusion matrices shown in Figure 5 are concerning. Matrices A and B just tells that the network has learned to classify everything as control, and this does not look right. Table 1 just tells the same for “no data” and “x-y flip, zoom” as the accuracy is 0.5 and specificity 1.0. The authors need to carefully look into this problem and train the network properly.

Histogram normalisation is not an augmentation strategy, as it seems to be more of a pre-processing step (as it is applied to both validation and testing datasets). This is confusing.   

Little evidence that the proposed network is the best model as no comparison study is provided. Other networks such as ResNet50 and VGG should have been used for comparison purposes (both are available in MATLAB).

Having said all the above, there is not enough evident to claim that “the phantom is ideal for AI applications”,

“for some training applications”, line 172 and 175, this is not clear. Which applications? Please clarify.

“porcine subject that had been recently euthanized from an unrelated Institutional Animal Care and Use Committee approved protocol was used”, this is not clear. Please confirm the use of animals was ethical.

 Minor comments:

This paper implies that US and POCUS is mainly used in emergency and military medicine, but this is not the case, as US is a widely used modality.

The following statement needs a reference: “The inclusion of ultrasonography specific modes, such as M-mode (still image motion over time), increases the number of methods a deep learning model can use to detect a pneumothorax using ultrasonography.”

 What is “initial training”? line 72.

“Learn rate” à learning rate

Reviewer 3 Report

In this paper, a pneumothorax (PTX) dynamic synthetic tissue model composed of synthetic gelatin model and lung model in custom 3D printed rib model is established, and the development of this model and its application in training image classification algorithm are introduced in detail. The authors used their previously developed deep learning image classification algorithm for shrapnel detection to accurately predict the presence of PTX in pig images by training only model image sets. The development of this synthetic phantom device solves the current gap in generating large image data sets for deep learning algorithm development. At the same time, the device is easy to produce, easy to modify to simulate different objects, and allows almost unlimited image acquisition, which will help to realize diagnosis automation using AI methods.

In general, the overall paper is well organized in theory, but there may be some minor problems in the format.

1. The picture is not clear enough. For example, the text clarity in Fig. 1 is inconsistent with that in Fig. 2, which means that Fig. 1 may not be the original but a screenshot.

2. The table format does not meet the requirements, and it should be a three-stage table.

3. The picture annotation format may need to be checked. It should be "figure 1 (a)" or "Fig 1(A)”.

4. The chapter subtitle is not very proper at the end of the page (such as the title of Section 2.5), and should be adjusted to the top of the page.

5. Picture centering problem: the picture should be centered on the text and should be carefully checked before publication.

In conclusion, attention should be paid to the details in the paper, which needs to be revised by the author before publication.
